# Clinical Impact of Preoperative Biliary Drainage in Patients with Ductal Adenocarcinoma of the Pancreatic Head

**DOI:** 10.3390/diagnostics13071281

**Published:** 2023-03-28

**Authors:** Maria João Amaral, João Freitas, Mariana Amaral, Marco Serôdio, Rui Caetano Oliveira, Paulo Donato, José Guilherme Tralhão

**Affiliations:** 1General Surgey Department, Centro Hospitalar e Universitário de Coimbra, 3000-075 Coimbra, Portugal; 2Faculty of Medicine, University of Coimbra, 3000-548 Coimbra, Portugal; 3Pathology Department, Centro Hospitalar e Universitário de Coimbra, 3000-075 Coimbra, Portugal; 4Clinical Academic Center of Coimbra (CACC), 3000-075 Coimbra, Portugal; 5Coimbra Institute for Clinical and Biomedical Research (iCBR) Area of Environment, Genetics and Oncobiology (CIMAGO), Faculty of Medicine, University of Coimbra, 3000-548 Coimbra, Portugal; 6Radiology Department, Centro Hospitalar e Universitário de Coimbra, 3000-075 Coimbra, Portugal; 7Biophysics Institute, Faculty of Medicine, University of Coimbra, 3000-548 Coimbra, Portugal

**Keywords:** pancreatic cancer, pancreaticoduodenectomy, obstructive jaundice, biliary drainage, prognosis

## Abstract

Our aim was to study the association between preoperative biliary drainage (PBD) and morbidity following cephalic pancreaticoduodenectomy (CPD) for pancreatic ductal adenocarcinoma (PDAC) and its prognostic impact, which is still controversial in the literature. A retrospective study was conducted, which included 128 patients who underwent CPD for PDAC, divided into two groups: those who underwent PBD (group 1) and those who did not undergo this procedure (group 2). Group 1 was subdivided according to the drainage route: endoscopic retrograde cholangiopancreatography (ERCP), group 1.1, and percutaneous transhepatic cholangiography (PTC), group 1.2. 34.4% of patients underwent PBD, and 47.7% developed PBD-related complications, with 37% in group 1.1 and 64.7% in group 1.2 (*p* = 0.074). There was a significant difference between group 1 and 2 regarding bacterial colonization of the bile (45.5% vs. 3.6%, *p* < 0.001), but no difference was found in the colonization by multidrug-resistant bacteria, the development of Clavien–Dindo ≥ III complications, clinically relevant pancreatic fistula and delayed gastric emptying (DGE), intra-abdominal abscess, hemorrhage, superficial surgical site infection (SSI), and readmission. Between groups 1.1 and 1.2, there was a significant difference in clinically relevant DGE (44.4% vs. 5.9%, *p* = 0.014) and Clavien–Dindo ≥ III complications (59.3% vs. 88.2%, *p* = 0.040). There were no significant differences in median overall survival and disease-free survival (DFS) between groups 1 and 2. Groups 1.1 and 1.2 had a significant difference in DFS (10 vs. 5 months, *p* = 0.017). In this group of patients, PBD was associated with increased bacterial colonization of the bile, without a significant increase in postoperative complications or influence in survival. ERCP seems to contribute to the development of clinically significant DGE. Patients undergoing PTC appear to have an early recurrence.

## 1. Introduction

Pancreatic cancer is among the neoplasms with the worst prognosis. Pancreatic ductal adenocarcinoma (PDAC) is the most common exocrine pancreatic neoplasm [1,2], having the highest case-fatality rate of any solid tumor [3] and representing the seventh leading cause of cancer-related death [1,2,4], with a 5-year overall survival (OS) of only 10% [5]. Patients with PDAC are often asymptomatic in the early stages of the disease [2,6,7], and only 15 to 20% of patients are resectable at diagnosis.

For PDAC located in the head of the pancreas, jaundice is the most common clinical sign at diagnosis. Progressive and prolonged obstructive jaundice leads to fatigue, malnutrition, bile stasis, and endotoxemia, being associated with hepatic dysfunction, coagulopathy, infections, anastomotic leakage, and delayed recovery after surgery [8]. Preoperative biliary drainage (PBD) aims to restore the normal bile flow. It is the only procedure that allows for non-resectable jaundiced patients to be treated with neoadjuvant therapy [9], and a biopsy can be performed in the same intervention in order to reach a definitive diagnosis [10]. Furthermore, it allows for resectable patients to safely wait for surgical resection when it cannot be performed in a short-term period when they develop severe pruritus, severe hyperbilirubinemia, acute cholangitis, or jaundice-related systemic complications [11,12]. This procedure is not routinely recommended for patients taken in for surgical resection, as it can increase complications [5], and formal indications for its use are still debated [8]. PBD is sometimes performed in the absence of the previously mentioned indications, which makes it difficult to assess and discuss its real impact on postoperative morbidity and patients’ prognosis.

Some studies have shown that PBD is associated with a decrease in the rate of postoperative complications [8,13], while others have shown that this would be the same whether or not patients underwent PBD [11,14,15]. However, what seems to be the increasing consensus is that it leads to an increase in the rate of postoperative complications, especially infectious ones [9,16,17,18,19,20,21,22,23,24,25,26,27]. The number of patients requiring this preoperative procedure is expected to rise due to the increasing use of neoadjuvant chemotherapy in pancreatic cancer patients [28]. Thus, the aim of this work was to clarify the association between PBD and complications following cephalic pancreaticoduodenectomy (CPD), and its prognostic impact.

## 2. Materials and Methods

### 2.1. Study Design

A retrospective study was conducted that included 128 consecutive patients who underwent CPD for PDAC, between January 2008 and August 2021 at our department, without neoadjuvant treatment. Routinely, a laparotomic cephalic pancreaticoduodenectomy without pyloric preservation and a standard lymph node dissection was the technique used. The pancreatic stump was managed with an end-to-side pancreaticojejunostomy (with or without duct stenting). PBD was performed as a “bridge” therapy, without pre-established protocols, in patients with obstructive jaundice with associated severe pruritus, cholangitis, comorbidities requiring preoperative work-up, and a lack of early access to surgery or in case of a patient’s willingness to postpone surgical intervention. Patients were divided into two groups: those who were submitted to PBD (group 1) and those who did not undergo this procedure (group 2). Group 1 was further divided according to the drainage route performed by endoscopic retrograde cholangiopancreatography (ERCP) (group 1.1) or by percutaneous transhepatic cholangiography (PTC) (group 1.2).

Preoperative analytical parameters included in the study were aspartate aminotransferase (AST), alanine aminotransferase (ALT), alkaline phosphatase (AP), gamma-glutamyltransferase (GGT), total bilirubin (BR), and albumin. Jaundice was considered for a BR value greater than 2.5 mg/dL [29]. Intraoperative variables considered were bile cultures (collected after sectioning the main bile duct), blood transfusions, and duration of surgical intervention. Regarding the postoperative period, variables considered were postoperative complications, namely postoperative pancreatic fistula (POPF), delayed gastric emptying (DGE), postoperative hemorrhage (POH) and surgical site infection, and readmission (until 30 days after surgery). Postoperative complications were defined as those occurring in the first 30 days after surgery and classified according to the Clavien–Dindo classification (CDC) [30]. A clinically significant postoperative complication was considered when patients met the criteria for CDC grade III or higher. DGE [31] and POPF [32] were defined according to the International Study Group of Pancreatic Surgery, considering grades B and C as clinically significant. Postoperative surgical site infection was divided into superficial surgical site infection (SSI) and deep surgical site infection, equivalent to an intra-abdominal abscess (IAA). Histopathological characteristics, relapse, OS, and disease-free survival (DFS) were also included. DFS was calculated from the date of surgery to the date of relapse and the OS from the date of surgery to the date of death or of data analysis. Preoperative risks for morbimortality were calculated using the American College of Surgeons (ACS) National Surgical Quality Improvement Program (NSQIP) Surgical Risk Calculator.

Data were obtained by reviewing the patients’ clinical histories, using the hospital database records. The study was approved by our Hospital’s Ethics Committee and was performed according to the Declaration of Helsinki [33].

### 2.2. Statistical Analysis

Statistical analysis was performed on IMB SPSS software, version 27.0 (IMB corporation, Armonk, NY, USA). First, a descriptive analysis was performed. Metric variables were presented by mean whenever there was a normal distribution and by the median if not. Relational statistics were performed using the chi-squared test or the Fisher exact test for qualitative variables and the Student’s *t*-test or Mann–Whitney U test for quantitative variables. Receiver operating characteristic (ROC) curves were used for BR cut-offs. The Kaplan–Meier and log-rank tests were used to conduct survival analysis, and univariate Cox regression was performed with the statistically significant variables from the survival analysis. In all the tests used, a *p*-value ≤ 0,05 was considered statistically significant.

## 3. Results

### 3.1. Preoperative Biliary Drainage

Of the 128 patients included in the study, 78 (60.9%) were male with a median age of 69 years (IQR 60.25–76). Forty-four (34.4%) patients underwent PBD before surgery while 65.6% did not. Preoperative AST, ALT, AP (*p* < 0.001), TB (*p* = 0.001), and GGT (*p* = 0.04) differed significantly between groups 1 and 2. Drainage was performed by PTC in 17 cases (38.6%) and by ERCP in 27 cases (61.7%). In 22 of those who underwent ERCP (81.5%), a plastic prosthesis was placed, and, in 2 cases (7.4%), a metallic one was used. In those who underwent PTC, an external biliary drain was used in 14 cases (82.4%), a mixed biliary drain was used in 1 case (5.9%), and a prosthesis was used in 2 cases (11.8%). The median drainage time was 30 days (IQR 20–46).

In 37% of patients in group 1.1 and in 64.7% of group 1.2, there was drainage-related morbidity (*p* = 0.074). Acute cholangitis occurred in 60% of patients in group 1.1, in 18.2% of patients with drainage-related morbidity in group 1.2 (*p* = 0.080), and acute pancreatitis in 30% and 0% of patients with drainage-related morbidity, respectively (*p* = 0.090). No patients in group 1.1 developed drainage-related bleeding, which occurred in 27.3% of patients with drainage-related morbidity in group 1.2 (*p* = 0.214). In group 1.2, no patient developed drainage obstruction, but this complication occurred in 20% of patients in group 1.1 (*p* = 0.214). Finally, drain mobilization occurred in 54.5% of patients with drainage-related morbidity in group 1.2. Results are detailed in the Appendix A.

### 3.2. Intraoperative Variables and Postoperative Complications

Intraoperative bile cultures were positive in 23 (18%) of the 128 patients, and 10 (43.5%) were positive for multidrug-resistant bacteria. Characteristics of intraoperative bile cultures are detailed in Table 1. Statistically significant differences were observed between groups 1 and 2 regarding the bacterial colonization of bile (45.5% vs. 3.6%, *p* < 0.001) but not regarding colonization by multidrug-resistant bacteria (45% vs. 33.3, *p* = 1.000). Between groups 1.1 and 1.2, there was a difference in bacterial colonization of bile, although without statistical significance (55.6% vs. 29.4%, *p* = 0.130).

Comparison between the postoperative complications of groups 1 and 2 and groups 1.1 and 1.2 are detailed in Appendix A, respectively. Between groups 1 and 2, no statistically significant differences were observed regarding postoperative morbidity (81.8% vs. 71.4%, *p* = 0.277) and mortality (4.5% vs. 11%, *p* = 0.223) in the development of clinically significant pancreatic fistula (16.7% vs. 15.6%, *p* = 0.878), clinically significant DGE (29.5% vs. 29.8%, *p* = 0.976), IAA (13.6% vs. 21.4%, *p* = 0.257), POH (15.9% vs. 10.7%, *p* = 0.428), SSI (6.8% vs. 10.7%, *p* = 0.539), readmission (50% vs. 48.8%, *p* = 1.000), and clinically significant postoperative complications (CCD ≥ III) (29.5% vs. 31%, *p* = 0.802). Overall, 45 patients (35.2%) required a postoperative blood transfusion, 47.7% of those undergoing PBD and 28.6% not undergoing this procedure (*p* = 0.031).

Between groups 1.1 and 1.2, there were statistically significant differences in the development of clinically significant DGE (44.4% vs. 5.9%, *p* = 0.014) and in clinically relevant postoperative complications (CCD ≥ III) (59.3% vs. 88.2%, *p* = 0.040), with no significant differences regarding postoperative morbidity (88.9% vs. 70.6%, *p* = 0.277) and mortality (3.7% vs. 5.9%, *p* = 1.000), the development of clinically significant pancreatic fistula (20% vs. 11.8%, *p* = 0.482), IAA (22.2% vs. 0%, *p* = 0.067), POH (18.5% vs. 11.8%, *p* = 0.689), SSI (11.1% vs. 0%, *p* = 0.272), and the readmission rate (51.9% vs. 47.1%, *p* = 0.757).

It was found that the risks of any complication (*p* = 0.020), readmission (*p* = 0.020), surgical reintervention (*p* = 0.034), death (*p* = 0.005), and sepsis (*p* = 0.025), calculated using the ACS NSQIP Surgical Risk Calculator, differed significantly between groups 1.1 and 1.2, with the risks being higher in group 1.2. On the other hand, the calculated risk of DGE was not significantly different between these groups (*p* = 0.695).

In jaundiced patients before surgery, a preoperative BR serum level of 17.65 mg/dL or higher (sensitivity 75%; specificity 67.5%) significantly discerned patients with SSI than those without, resulting in an area under the curve (AUC) of 0.724 (95%CI 0.558–0.889, *p* = 0.037) (Figure 1).

ROC curve analysis did not show statistically significant results for BR serum cut-off values to predict postoperative morbidity in general (AUC 0.389, 95%CI 0.251–0.527, *p* = 0.118), CDC ≥ III (AUC 0.552, 95%CI 0.421–0.683, *p* = 0.431), postoperative mortality (AUC 0.414, 95%CI 0.200–0.627, *p* = 0.482), POH (AUC 0.437, 95%CI 0.245–0.629, *p* = 0.499), IAA (AUC 0.514, 95%CI 0.377–0.652, *p* = 0.848), clinically significant POPF (AUC 0.497, 95%CI 0.253–0.740, *p* = 0.977), clinically significant DGE (AUC 0.500, 95%CI 0.376–0.624, *p* = 0.997), and readmission (AUC 0.533, 95%CI 0.415–0.651, *p* = 0.583).

### 3.3. Histopathological Characteristics

The median tumor size was 2.55 cm for group 1 (IQR 2.4–3.5) and 3.2 cm (IQR 2.675–4) for group 2 (*p* = 0.036). There was a significant difference in lymphovascular invasion between the two groups (90.9% vs. 75%, *p* = 0.031); however, there were no significant differences in tumor staging (*p* = 0.501), lymph node invasion (81.8% vs. 76.2%, *p* = 0.464), perineural invasion (95.5% vs. 88.1%, *p* = 0.217), and in positive surgical resection margins (50% vs. 53.6%, *p* = 0.701). See Appendix A for detailed results.

### 3.4. Follow-up and Survival

The median follow-up time for all patients was 15.5 months (IQR 7.25–30.75), 15 months for group 1 and 16 months for group 2. For group 1.1, the median follow-up time was 17 months, and, for group 1.2, it was 13 months. The median OS was 18 months, and the OS survival rate at 3 and 5 years was 28.6% and 18.5%, respectively. For groups 1 and 2, the median OS was 20 and 18 months (*p* = 0.833), and, for groups 1.1 and 1.2, it was 24 and 14 months (*p* = 0.258), respectively. The OS rate at 5 years was 18.1% and 18.5% for groups 1 and 2, and it was 18.2% and 18.5% for groups 1.1 and 1.2, respectively.

Of the 128 patients, 70 (61.9%) had a relapse by the time of analysis, but no statistically significant differences were found between the relapse rate of groups 1 and 2 (66.7% vs. 59.5%, *p* = 0.453) and groups 1.1 and 1.2 (66.7% in each group, *p* = 1.000). Of the 70 patients, 42 (60%) had hepatic recurrence, 29 (41.4%) had local recurrence, 29 (41.4%) had pulmonary recurrence, and 14 (20%) has peritoneal recurrence. Regarding pulmonary recurrence, groups 1 and 2 differed significantly (57.7% vs. 31.8%, *p* = 0.048), with no significant differences in local recurrence (50% vs. 36.4%, *p* = 0.297), hepatic recurrence (57.7% vs. 61.4%, *p* = 0.587), and peritoneal recurrence (19.1% vs. 20.5%, *p* = 0.828). Results for groups 1.1 and 1.2 were as follows: local recurrence (50% vs. 50%, *p* = 1.000), pulmonary recurrence (68.8% vs. 40%, *p* = 0.228), hepatic recurrence (43.8% vs. 80%, *p* = 1.09), and peritoneal recurrence (12.5% vs. 30%, *p* = 0.340).

Median DFS was 9 months, with a 3-year DFS rate of 2.9% and a 5-year DFS rate of 1.4%. For groups 1 and 2, the median DFS was 10 and 8 months (*p* = 0.192), and, for groups 1.1 and 1.2, it was 10 and 5 months (*p* = 0.017), respectively (Figure 2). The DFS rate at 5 years was 3.8% and 0% for groups 1 and 2 and 6.3% and 0% for groups 1.1 and 1.2, respectively. Univariate regression showed that patients in group 1.2 had a 2.559-fold increased risk of having an early recurrence (HR 2.559, 95%IC 1.092–5.994, *p* = 0.031).

## 4. Discussion

CPD is the only curative treatment for cephalic PDAC, although it still has high postoperative morbidity and mortality. Prevention of complications following this procedure is very important since these can decrease access to adjuvant therapy and, consequently, increase recurrence and decrease OS [34].

In our cohort of 128 patients, 34.4% underwent PBD, mostly by ERCP (61.7%). ERCP and PTC for PBD were compared in a randomized trial, and higher success rates and lower complications were seen in the endoscopic group (19 vs. 67%) [12], which agrees with our study, where there was a 37% drainage-related complication rate in group 1.1 and 64.7% in group 1.2 (*p* = 0.074). On the contrary, a metanalysis found lower procedure-related complication rates in the percutaneous group (OR = 44, 95%CI 0.23–0.84, *p* = 0.01) [35], and another study found similar complication rates between the two procedures [36]. These studies do not include only patients with PDAC, which is contrary to ours.

The bile cultures were positive in 45.5% of the patients submitted to PBD and in only 3.6% of patients who did not undergo PBD (*p* < 0.001), but we could not find significant differences in multidrug-resistant bacteria positivity between the two groups. Patients submitted to ERCP had a higher positivity rate than those undergoing PTC (55.6% vs. 29.4%, *p* = 0.130). A prospective cohort study confirmed the high incidence of positive bile cultures in patients submitted to PBD (98.2% vs. 25.9%, *p* < 0.001) and that the performance of ERCP increases this risk. Moreover, it showed the lack of correlation between contamination of intraoperative bile cultures and abdominal infectious complications and that the complete concordance of bile and organ space infections is low [37], which probably could make us reconsider the need for routine performance of intraoperative bile cultures.

Performing PBD did not influence postoperative morbidity and mortality in our study. Two metanalyses concluded that PBD increases postoperative morbidity without influencing postoperative mortality [16,21]. Likewise, in our study, there was no significant association with the development of clinically significant POPF, POH, and IAA, which are results that agree with the literature [9,16,21,22,25], nor with the development of SSI, a discordant result [9,16,21,22,25,38]. Suragul et al. identified PBD as an independent risk procedure for the development of SSI after pancreaticoduodenectomy (OR 3.04, 95%CI 1.36–6.79, *p* < 0.05) but not for organ/space infection (IAA) [27]. In our study, PBD also did not influence the development of clinically significant DGE. The literature is more controversial regarding this subject, as one metanalysis reported an increased incidence of PBD-related DGE (OR 1.21, 95%CI 1.03–1.42, *p* = 0.02) [16], while other studies stated the opposite [9,22,25].

Regarding perioperative blood transfusions, the study by Ray et al. showed no significant differences whether patients underwent PBD or not (25.7% vs. 21.8%, *p* = 0.408) [25]. Mezhir et al. obtained the same results, however, and found that patients undergoing PBD lost more blood during the surgical procedure [38], and Santos et al. reported that intraoperative bleeding was less significant in patients submitted to PBD [11]. In our study, although we had no records of the amount of blood lost during the surgical procedures, it was found that PBD was associated with a greater need for blood transfusions (47.7% vs. 28.6%, *p* = 0.031). This result is controversial yet supported by other studies [39,40]. This group of patients can have transient hemobilia as post-PBD bleeding [41]. Perioperative blood transfusions have been shown to reduce patients’ OS [42,43] and seem to be a prognostic factor in PDAC patients undergoing surgical resection [39,40].

This study found a significant association between PBD and lymphovascular invasion, which is a well-characterized, independent prognostic factor for PDAC [44,45,46]. A study by Ahn et al. assessed the impact of PBD on the prognosis of the ampulla of Vater carcinoma patients and, despite not having statistically significant results, found that 79.5% of patients undergoing ERCP, compared to 58.8% of non-drained patients, had lymphovascular invasion [47]. However, the association between PBD and lymphovascular invasion in PDAC is, to our knowledge, still not studied in the literature.

Using the ACS NSQIP Surgical Risk Calculator, patients drained using PTC had a preoperative higher probability of postoperative morbidity, sepsis, death, and readmission. However, our study did not show significant differences in postoperative morbidity, mortality, readmission, grade B or C POPF, IAA, POH, or SSI. Patients submitted to PTC (group 1.2) had a significant increase in CDC ≥ III (59.3% vs. 88.2%, *p* = 0.040). Those undergoing PBD by ERCP (group 1.1) developed clinically significant DGE more frequently (44.4% vs. 5.9%, *p* = 0.014), although the preoperative probability of this complication was not significantly higher according to the ACS NSQIP Surgical Risk Calculator. As previously reported by Wu et al. [48], these patients had an increase in the rate of IAA in our study, although this was not statistically significant. In a metanalysis by Dorcaratto et al. the drainage route was found to be unrelated to the development of clinically significant postoperative complications, POPF, SSI, and postoperative mortality; however, the PTC group had fewer overall postoperative morbidity [35]. El-Haddad et al. reported that the drainage route was unrelated to the development of SSI, POH, DGE, POPF, and postoperative mortality [49].

Our results showed that PBD does not significantly influence OS and DFS. These results agree with some studies [23,50,51,52,53] and disagree with others [39,54]. Interestingly, although it does not influence DFS or recurrence rates overall, it was found that patients undergoing PBD have higher rates of pulmonary recurrence (57.7% vs. 31.8%, *p* = 0.048). Studies show that the lung is the most common metastatic site among long-term survivors with PDAC [55,56]. Metastatic organotropism to the lung might be related, among other factors, to immune features (like an inflammatory phenotype) [57]. We hypothesize that biliary drainage causes inflammation that could promote tumor progression and metastasis [58].

Likewise, there was no significant difference in the recurrence rates for patients undergoing PBD by ERCP or PTC. However, patients drained using PTC had a significantly lower DFS compared to patients undergoing ERCP (10 vs. 5 months, *p* = 0.017), contrary to some reports [53], without a significant difference in OS (24 vs. 14 months, *p* = 0.258), which disagrees with some studies [39,40,59] but agrees with another [50]. Those submitted to PTC also had higher rates of hepatic and peritoneal recurrence, which may be due to the procedure itself, because, during its performance, it can cause intrahepatic and peritoneal dissemination of neoplastic cells, as shown previously [60]. Although without statistical significance in our study, this result may have an important clinical impact and agrees with the literature [39,40,59,60].

We are aware of some limitations of this study. First, its retrospective and unicentric design can induce some biases, particularly limited control in obtaining the patients’ sample and consulting patient’s records, the fact that some patients may have undergone PBD outside our hospital and even before evaluation by a surgeon, and the lack of clear indications for PBD and the drainage route. Second, we analyzed preoperative biological data but not the delay between PBD and blood tests, and our database did not include the duration of preoperative jaundice or BR level before PBD. In addition, due to the low number of patients, we could not specify our results according to the type of drainage material, type of biliary stent (plastic stent vs. self-expandable metal stent), or type of drain (external drain or mixed drain). Finally, the fact that the sample size was small and included patients over a long period of time, during which surgical techniques, perioperative care, and adjuvant therapies have evolved, can influence the results.

In conclusion, in this sample of PDAC patients, PBD is associated with a significant increase in the bacterial colonization of bile, without a significant rise in postoperative morbidity and mortality or influence on survival. ERCP increases bile colonization and seems to contribute to a clinically relevant DGE after CPD. Patients undergoing PBD by PTC have an earlier recurrence and have higher rates of hepatic and peritoneal recurrence. For these reasons, a clear definition of PBD indications is essential.

## Figures and Tables

**Figure 1 diagnostics-13-01281-f001:**
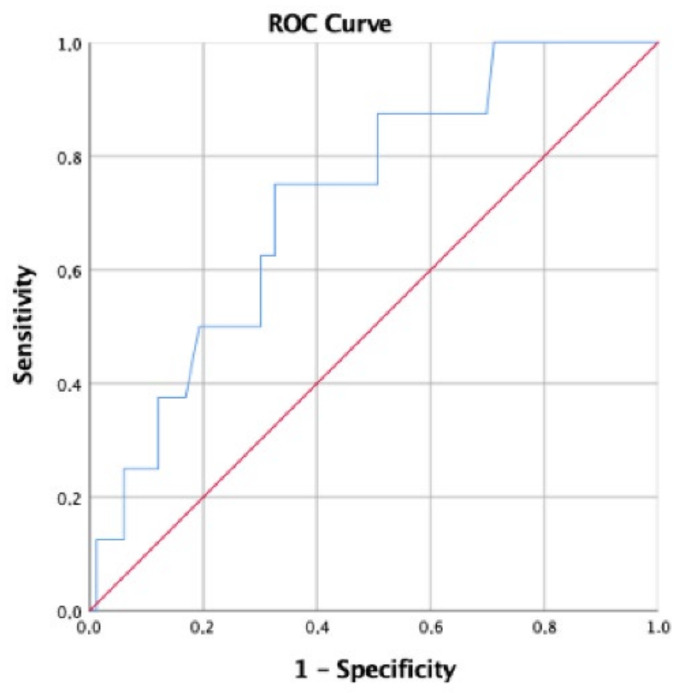
ROC curve for preoperative BR serum level and SSI.

**Figure 2 diagnostics-13-01281-f002:**
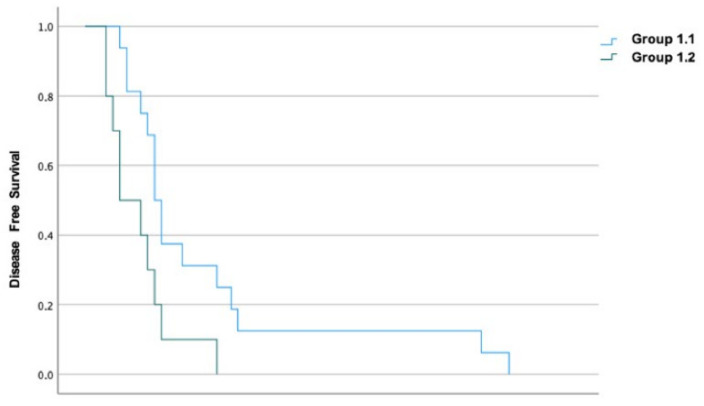
Kaplan–Meier curves for DFS in groups 1.1 and 1.2.

**Table 1 diagnostics-13-01281-t001:** Detailed intraoperative bile cultures.

	Group 1 *n* = 44 (%)	Group 2 *n* = 84 (%)	Total *n* = 128 (%)	*p*
Positive bile culture	20 (45.5)	3 (3.6)	23 (18)	<0.001
Positive for multidrug-resistant bacteria	9 (45)	1 (33.3)	10 (43.5)	1.000
Monomicrobial cultures	14 (70)	3 (100)	17 (73.9)	0.539
Polymicrobial cultures	6 (30)	0	6 (26.1)	0.539
*Escherichia coli*	2 (10)	2 (66.7)	4 (17.4)	0.067
*Klebsiella pneumoniae*	3 (15)	0	3 (13)	1.000
*Enterobacter clocae*	3 (15)	0	3 (13)	1.000
*Enterococcus faecium*	5 (25)	0	5 (21.7)	1.000
Others	11 (55)	1 (33.3)	12 (52.2)	0.590

## Data Availability

All data generated or analyzed during this study are included in this published article (and its Appendix A).

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
