# Peer review of "Clinical Impact of Preoperative Biliary Drainage in Patients with Ductal Adenocarcinoma of the Pancreatic Head"

_diagnostics, 2023, doi:10.3390/diagnostics13071281_

Round 1

Reviewer 1 Report

This is a retrospective study investigating a still debated issue: the preoperative biliary drainage before pancreatic surgery. The manuscript is well-organized and written.

Please explain the reasons for performing 38.6% of biliary drainage by PTC instead of ERCP. 

The importance of preoperative diagnosis, especially in the era of neoadjuvant chemotherapy should be mentioned. Doing so, cite PMID: 35915956

Author Response

Thank you for your review and comment on our paper. Changes to the manuscript are shown using the “track changes” function.

There is a 38.6% rate of drainage by PTC due to the fact that, in our hospital, this option is available 24 hours a day, 365 days a year. That it not the case for ERCP, we don’t have access to it that easily.

Reviewer 2 Report

Dear Collegues, your paper could me interesting, unfortunately there is some bias that reduce clinical improvement.

Infact it's not clear why patiens receive preoperative drainage or not and a stratification of data made on biomarkers .

The risk of biliary bacterial contamination in ERCP patients is quite obvious for the impact of technique on intrumented billiary tract.

It's not clear if you performed a somastathine or similar drug protection before intervention, after or only in case of bilio digestive lack.

So your work could be interesting if give us some suggestion to choose when billiary drainage could be useful or not.

In italy we prefer to not place drainage if clinical situation is stable because we prefer work on large biliary duct.

Author Response

Thank you for your review and comments on our paper. Changes to the manuscript are shown using the “track changes” function.

Patients were submitted to preoperative biliary drainage in case of obstructive jaundice together with cholangitis, severe pruritus, comorbidities requiring preoperative work-up, lack of early access to surgery or the patient’s willingness to postpone the intervention.

In our department, octeotride is usually given to patients during surgery and in the days after surgery.

Round 2

Reviewer 2 Report

The modifications done make paper more clear and give information that can help surgeon to choose, without any obstacle, if drain biliary duct or not in patient with pancreatic cancer and jaundice.

My opinion is that your work reinforce the necessity to avoid biliary leakage during surgical intervention and in postoperative period especially in patients with preoperative drainage due to high risk of violent sepsis.

Thank you